# Investigation of the Effect of Volumetric Hydrophobization on the Kinetics of Mass Transfer Processes Occurring in Cement Concretes during Corrosion

**DOI:** 10.3390/ma16103827

**Published:** 2023-05-18

**Authors:** Viktoriya S. Konovalova

**Affiliations:** Department of Natural Sciences and Technosphere Safety, Ivanovo State Polytechnic University, Sheremetevsky Ave., 21, Ivanovo 153000, Russia; kotprotiv@yandex.ru

**Keywords:** volumetric hydrophobization, corrosive mass transfer, concrete corrosion, corrosion kinetics, hydrophobizing additive, hydrophobized concrete, waterproof concrete

## Abstract

The entry of aggressive substances into the pore structure of concrete causes the development of corrosion processes and leads to the destruction of the cement stone structure. Hydrophobic additives provide high density and low permeability and are an effective barrier to the penetration of aggressive substances into the structure of cement stone. To assess the contribution of hydrophobization to the durability of the structure, it is necessary to know how much the processes of corrosive mass transfer slow down. To study the properties, structure and composition of the materials studied in the work (solid and liquid phases) before and after exposure to liquid-aggressive media, experimental studies were carried out using chemical and physicochemical analysis methods: determination of density, water absorption, porosity, water absorption and strength of cement stone; differential thermal analysis; quantitative analysis of calcium cations in liquid medium by complexometric titration. The article presents the results of studies of the effect on the operational characteristics of the introduction of a hydrophobic additive of calcium stearate into the cement mixture at the stage of concrete production. The effectiveness of volumetric hydrophobization was evaluated to prevent the penetration of an aggressive chloride-containing medium into the pore structure of concrete destruction and the leaching of calcium-containing components of cement stone. It was found that the introduction of calcium stearate in an amount of 0.8–1.3% by weight of cement increases the service life of a concrete product during corrosion in liquid chloride-containing media with a high degree of aggressiveness by four times.

## 1. Introduction

During the operation of concrete, various aggressive substances affect it, which leads to a decrease in its durability. The durability assessment is related to the expected performance characteristics of the product and consists in determining the factors that are crucial for the destruction of the material depending on its composition and properties, determining changes caused by the interaction of the material with external-aggressive media, and measuring damage over time.

The main physicochemical processes of destruction in reinforced concrete products occur due to the penetration of water into the concrete structure [1,2,3,4,5]. The penetration of water into the pores occurs by various mechanisms and is determined by the degree of saturation of concrete. The microstructure of concrete has a great influence on the rate of water penetration. When unsaturated concrete is exposed to an aqueous environment, even in the absence of pressure, water will be absorbed into the pore structure due to capillary suction [6,7].

In order to prevent and reduce the penetration of water and dissolved aggressive substances into the pore structure of concrete, its hydrophobization is ensured. The purpose of volumetric hydrophobization is to compact concrete to prevent water ingress and/or to make the capillary surface non-wettable, which will eventually reduce water penetration.

Special additives are used to create waterproof concrete. They reduce the probability of water ingress, reducing the permeability and shrinkage of concrete during drying [8]. Additives are introduced into the water-cement mixture in small quantities to increase the durability of concrete, improve its properties, and control setting or hardening [9,10,11]. It can be liquid or powdery substances. Hydrophobizing additives differ in chemical basis depending on the type of concrete, cement mortar, or plaster [12].

Hydrophobic additives introduced into concrete usually act on the basis of one or a combination of three mechanisms: reduction of capillarity by reducing the water–cement ratio, hydrophobization of capillaries, and physical or chemical blocking of pores. Reactive silicates [13,14], calcined clays [15,16], colloidal silicon dioxide [17,18], lignosulfonates [19], naphthalene sulfonate formaldehyde [20,21,22], sulfated melamine formaldehyde [22,23], and polycarboxylate ether [24,25,26] are used to reduce the capillarity of concrete. Hydrophobization of capillaries is achieved by introducing into concrete soap [27], bitumen [28], mineral oils [29,30], fatty acids [31], calcium and zinc stearates [32,33,34,35], butyl stearate [36,37], acrylic resins [38,39], thin wax emulsions [40,41], silicones [42,43], silane-siloxanes [44,45]. For example, stearates react chemically with calcium hydroxide in cement, resulting in the formation of a layer of insoluble calcium stearate on the walls of the pores of cement stone, providing water repellency [46]. Physical pore blocking can be achieved with the help of inert powder fillers such as talc [47], bentonite [48] or with the help of fine waxes [49], bitumen [50], acrylic emulsions [51], and emulsion styrene–butadiene rubbers [52,53]. Chemical pore blocking is carried out using microfine latent hydraulic reactive silicates [54] or pozzolan nanoparticles [55,56], providing secondary hydration to achieve the most dense packing of C-S-H gel in a cement matrix at the nanoscale. In the above materials, available free lime Ca(OH)_2_ is used for recrystallization and formation of new, more stable C-S-H and C-A-S-H phases in the presence of water [57]. These materials used as additives are also called crystal additives.

The positive effect of concrete hydrophobization usually lies in the fact that this type of treatment prolongs the period before the onset of corrosion. When corrosion begins, the hydrophobicity of the surface of the pores of the cement stone effectively prevents the penetration of the liquid medium and reduces the rate of corrosion processes [58]. When examining the motorway pier after seven years of operation, no chlorides were found in the hydrophobized concrete [59]. Hydrophobic additives do not have a negative effect on the strength of concrete; on the contrary, some concretes with hydrophobic additives have increased strength [9,60]. Since water-repellent substances are evenly distributed over the entire volume of concrete, they do not change the appearance of the product. In addition, hydrophobic additives are effective for reducing efflorescence since the migration of water throughout the entire volume of concrete is reduced [61,62].

In individual and mass construction of structures operated under the influence of liquid media, the use of concrete grades for water resistance W10–W16 is widespread. The corrosion resistance of reinforced concrete with hydrophobic additives in a highly aggressive chloride-containing environment needs additional research. In order to assess the contribution of hydrophobization to the durability of a concrete structure, it is necessary to know how effectively this treatment method prevents the penetration of various aggressive substances into the structure of cement stone and how long this efficiency can be maintained.

## 2. Materials and Methods

### 2.1. Materials

For the production of cement stone samples, Portland cement with a standardized composition without mineral additives of the CEM I 42.5 N brand was used as a binder. The choice of this brand of Portland cement is due to its prevalence and demand in mass construction. The chemical and mineralogical composition of Portland cement of the CEM I 42.5 N grade, established by the quality certificate, are presented in Table 1 and Table 2.

Cement stone samples were made from solutions of normal density with a water–cement ratio W/C = 0.3. The tests were carried out after 28-day hardening of the samples at a temperature of 20 ± 2 °C and relative humidity of 50–70%. Corrosion studies were carried out on cubic samples of cement stone with a face of 3 cm. The cement stone was placed for 150 days in containers filled with a 2% MgCl_2_ solution. The choice of a 2% MgCl_2_ solution for corrosion tests is due to the high activity of chloride ions that create an aggressive environment for concrete [63]. The solutions were prepared and operated at a temperature of 20 °C.

The grade of cement for water resistance was regulated by a hydrophobic additive calcium stearate and, before the experiment, was determined by the method described in Section 2.2. Calcium stearate was introduced into the cement mixture at the stage of sample production to ensure volumetric hydrophobization. It was experimentally found that the grade of concrete W10 corresponds to the concentration of the hydrophobizer in the amount of 0.8% by weight of cement, grade of concrete W14–1.1%, and grade of concrete W16–1.3%. The characteristics of cement stone samples with calcium stearate additives are presented in Table 3.

Each type of test described below in this section was performed on a series of five cement stone samples. The results in the article are given for average values.

### 2.2. Method of Determining the Grade of Concrete by Water Resistance

The grade of concrete for water resistance was determined according to the method described in the patent [64] and MI 2625-2000 «Recommended Practice—National Measurement Standards—Cement Materials—Accelerated Method for Measuring Water Impermeability» [65]. After drying at a temperature of 105 ± 5 °C to a constant mass, the cement samples under study were split into two parts, after which paraffin waterproofing was applied to the side surfaces. Then, samples were saturated with water through the split surface in 1 and 5 min, respectively, and the equivalent capillary pressure in the material was found depending on the capillary porosity of the material and the volume of absorbed water. An indicator of the degree of the kinetics of water saturation was established, the volume of capillary pores of the mortar part of the material was determined, and the water resistance of cement materials was calculated by the formula:(1)$$\lambda^{\prime }={\left(\frac{P_{i}}{0.2}\right)}^{n}\cdot \sqrt{1+\frac{P_{i}}{P_{C}}},$$
where *P_i_* is definable water resistance [MPa], *P_C_* is equivalent capillary pressure resulting from water saturation of materials [MPa], and *n* is the indicator of the degree of the kinetics of water saturation of the material, determined by the formula:(2)$$n=\frac{lg\Delta {\bar{m}}_{5}-lg\Delta {\bar{m}}_{1}}{0.7},$$
where Δ$\bar{m}$_1_ and Δ$\bar{m}$_5_ are arithmetic mean values of the mass difference of water-saturated and dried samples for 1 and 5 min, respectively; *λ* is the criterion determined by the formula:(3)$$\lambda =\frac{V_{Ci}}{{\left(\tau^{\prime }\right)}^{n}\Delta V_{C1}},$$
where *V_Ci_* is the volume of capillary pores of the material in the test sample, [cm^3^]; Δ*V_C_*_1_ is the volume of water absorbed by the capillaries of the material in 1 min, [cm^3^]; *τ*′ is relative reduced time, depending on the size and shape of the tested samples (from MI 2625-2000).

The value of Δ*V*_*C*1_ is calculated by the formula:(4)$$\Delta V_{C_{1}}=\frac{\Delta {\bar{m}}_{1}}{\gamma_{L}}\cdot \left(1-\frac{K_{6}\cdot \Delta V^{\prime }C}{W-\beta_{1}S-\beta_{2}L-K_{3}\Delta V^{\prime }C}\right),$$
where *W* is the volume of the mixing water in 1 L of the compacted mixture of the material, [cm^3^]; *S*, *L* are the content by weight of fillers in 1 L of a compacted mixture of material, respectively small and large, [g]; *β*_1_, *β*_2_ are water absorption of aggregates in fractions of their mass during mixing and compaction of the mixture, respectively, small and large, [cm^3^/g]; *K*_3_, *K*_6_ are coefficients from MI 2625-2000; Δ*V*′ is the specific current contraction of the cement used at the time of testing the material (Δ*V*′ = 2486 cm^3^/g); *C* is the content by weight of cement in 1 L of compacted mixture, [g].

The equivalent capillary pressure *P_C_* is calculated by the formula:(5)$$P_{C}=\frac{A_{1}\cdot p_{cap}^{2}}{M\cdot \Delta V_{C_{1}}^{2}\cdot f^{2}},$$
where *A*_1_ is constant from in MI 2625-2000; *f* is the coefficient reflecting the ratio of the volume of concrete to the volume of its mortar part; *M*_1_ is the parameter that integrally reflects the features of the capillary structure and the «overflow» effect is calculated by the formulas:
–for concrete with a large aggregate: *M* = 6 + 300$p_{cap}^{3/2}$;–for fine-grained concrete: *M* = 12 + 540$p_{cap}^{3/2}$.

The volume of capillary pores in the material of the tested concrete sample is determined by the formula:(6)$$V_{C}=\frac{V\cdot p_{cap}}{f},$$
where *V* is the volume of the test sample, [cm^3^].

The coefficient *f* is determined by the formula:(7)$$f=\frac{1000\gamma_{L}}{1000\gamma_{L}-L}.\;$$

The capillary porosity of the solution part is determined by the formula:(8)pcap=W−β1S−β2L−K5ΔV′C1000−LγL, 
where *γ_L_* is the true density of coarse aggregate, [g/cm^3^]; K_5_ is the stoichiometric constant from MI 2625-2000.

The tested sample was assigned a water resistance mark equal to the lower closest to *P_i_* value W given in GOST 12730.5-2018 «Concretes. Methods for determination of water tightness» [66].

### 2.3. Determination of Density, Water Absorption, and Porosity of Concrete

Concrete density is defined as the ratio of the mass of concrete (sample) in a state of natural humidity to its total volume. The volume of samples of the correct shape is calculated by their geometric dimensions. The dimensions of the samples are determined by a ruler or a caliper with an error of no more than 1%. The mass of the samples is determined with an error of no more than 0.1%.

The density of concrete *ρ_c_* is calculated by the formula:(9)$$\rho_{c}=\frac{m}{V}\cdot 1000,\;$$
where *m* is sample weight, [g] and *V* is sample volume, [cm^3^].

The true density of concrete is determined by measuring the mass of the unit volume of the crushed and dried sample. The initial sample is crushed in a laboratory crusher to a size of less than 5 mm, mixed, and reduced to 150 g. The resulting sample is again crushed to a size of less than 1.25 mm, mixed, and reduced to 30 g. Then it is crushed in a mortar to a powdery state (less than 0.125 mm), dried to a constant mass, cooled to room temperature, and two samples of 10 g each are weighed.

Each sample is poured into a clean, dry pycnometer, where distilled water at room temperature is poured in such an amount that the pycnometer is filled with no more than half of its volume. The pycnometer is placed in a slightly inclined position on a sand or water bath; its contents are boiled for 20 min to remove air bubbles. After removing the air, the pycnometer is wiped, cooled to room temperature, filled to the mark with distilled water, and weighed. Then the pycnometer is released from the contents, washed, filled to the mark with distilled water at room temperature, and weighed. 

The true density of concrete *ρ* is calculated by the formula:(10)$$\rho =\frac{m\cdot \rho_{w}}{m+m_{1}-m_{2}},\;$$
where *m* is the mass of the crushed sample, [kg], *ρ_w_* is water density, [kg/m^3^], *m*_1_ is the weight of a pycnometer with distilled water, [kg], and *m*_2_ is the mass of the pycnometer with a sample and distilled water, [kg].

Water absorption is determined by testing a series of samples with dimensions 10 × 10 × 10 cm. The samples are placed in a container filled with water at a temperature of 20 ± 2 °C above the upper level of the stacked samples by about 50 mm. Samples are weighed every 24 h on a scale with an error of no more than 0.1%. When weighing, the samples are pre-wiped with a wrung-out damp cloth. The mass of water flowing out of the pores of the sample on the scale cup is included in the mass of the saturated sample. The test is carried out until the results of two consecutive weight measurements differ by no more than 0.1%.

The water absorption of concrete of a separate sample *W_m_* by weight as a percentage with an error of up to 0.1% is calculated by the formula:(11)$$W_{m}=\frac{m_{s}-m_{w}}{m_{s}}\cdot 100\%,\;$$
where *m_c_* is the mass of the dried sample, [g], and *m_w_* is the mass of the water-saturated sample, [g].

The water absorption of concrete of a separate sample *W*_0_ by volume as a percentage with an error of up to 0.1% is calculated by the formula:(12)$$W_{0}=\frac{W_{m}\cdot \rho_{0}}{\rho_{w}},$$
where *ρ*_0_ is the average density of dry concrete, [g/cm^3^] and *ρ_w_* is the density of water assumed to be equal to 1, [g/cm^3^].

The total pore volume of *V_p_* concrete is calculated as a percentage with an error of up to 0.1% according to the formula:(13)$$V_{p}=\frac{\rho -\rho_{d}}{\rho }\cdot 100\%,\;$$
where *ρ* is the true density of concrete, [kg/m^3^] and *ρ_d_* is the density of dried concrete, [kg/m^3^].

The volume of open capillary pores of concrete (sample) *V_o_* is assumed to be equal to the water absorption of concrete by volume *W*_0_.

The volume of conditionally closed capillary pores of concrete *V_c_* is calculated by the formula:*V_c_* = *V_p_* − *V_o_*.(14)

### 2.4. Determination of the Compressive Strength of Cement Stone

The strength was determined using a hydraulic press P-50 on samples of cement stone with a face length of 10 cm. The working area of the press consists of two horizontal plates mounted on a column with a lead screw. During the compression test, the samples were installed on the lower base plate of the press centrally relative to its longitudinal axis.

After installing the test sample on the base plates of the press (additional steel plates), the upper plate was combined with the upper support face of the sample (additional steel plate) so that their planes were completely adjacent to each other. Then the loading began.

The loading of the samples was carried out continuously with a constant rate of increase of the load until their destruction. At the same time, the loading time of the test sample before its destruction was at least 30 s. The maximum force achieved during the test was taken as a destructive load.

The compressive strength of concrete *R_c_* for each sample is calculated by the formula:(15)$$R_{c}=\alpha \frac{F}{A},$$
where *α* is the scale factor for reducing the strength of concrete to the strength of concrete in samples of basic size and shape, assumed to be 0.95 according to GOST 10180-2012 «Concretes. Methods for strength determination using reference specimens» [67]; *F* is destructive load, [N]; *A* is the area of the working section of the sample, [mm^2^].

### 2.5. Differential Thermal Analysis of Cement Stone

Differential thermal analysis (DTA) is a method of studying physical and chemical transformations accompanied by the release or absorption of heat. The method is used to register phase transformations in a sample and study their parameters. The analysis was carried out according to the standard method on the Q-1500D derivatograph.

In preparation for the study, the cement stone is crushed to pieces of 1-3 mm in size. The weight of crushed cement stone weighing 1–2 g is transferred to the box, filled with 100% ethyl alcohol, and left for 10 h. To separate the material from alcohol, filtration is carried out through a paper filter. The dehydrated material is crushed in a mortar to powder. Then, 0.5–1 g is taken from the powdered material, which is placed in the crucible of the derivatograph.

At a heating rate of 10 °C/min and a temperature range of 20–1000 °C, the temperature curve and the mass loss curve of the substance are fixed. The analysis of the derivatograms was carried out according to the standard methodology.

### 2.6. Quantitative Analysis of Calcium Ions in a Liquid Medium by the Method of Complexometry

The complexometric determination of calcium cations is based on direct titration of the test solution with a standard solution of complexon III with the addition of indicators of murexide or dark blue chromogen. The indicators form a complex compound of red color with calcium ions. When titrating the test solution with complexon III at the equivalence point, the red color changes to the color characteristic of the free indicator.

A total of 25 mL of the analyzed solution is pipetted into a conical flask, and 50 mL of distilled water, 25 mL of a 20% sodium hydroxide solution, and 2–3 drops of the indicator are added to it. Then the test solution is titrated with a 0.1 N solution of complexon III with continuous stirring until the red color changes to purple or blue. In the end, titration is carried out very slowly.

The calculation of the content of calcium cations in the solution is carried out according to the formula:(16)$$g_{A}=\frac{E_{A}\cdot N_{B}\cdot V_{B}}{1000},\;$$
where *g_A_* is the total content of calcium cations in solution, [g]; *E_A_* is the equivalent of calcium cations; *N_B_* is the concentration of standard EDTA solution, [eq/L]; *V_B_* is the volume of standard EDTA solution spent on titration, [mL].

### 2.7. Mathematical Model of the Second Type of Corrosion of Cement Concretes

When studying the II type of corrosion of cement concretes by our scientific school for the system under consideration, «cement stone–aggressive component», the mass transfer equations for calcium hydroxide are presented in the form of the following boundary value task [68]:(17)$$\frac{\partial C\left(x,\tau \right)}{\partial \tau }=k\frac{\partial^{2}C\left(x,\tau \right)}{\partial x^{2}}+\frac{q_{v}\left(x\right)}{\rho_{con}},\;\tau \geq 0,\;0\leq x\leq \delta ,\;$$
(18)$${\left.C\left(x,\tau \right)\right|}_{\tau =0}=C_{0}\left(x\right),$$
(19)$${\left.\frac{\partial C\left(x,\tau \right)}{\partial x}\right|}_{x=0}=0,$$
(20)$$-k\rho_{б}{\left.\frac{\partial C\left(x,\tau \right)}{\partial x}\right|}_{x=\delta }=q_{n}.$$
where *C*(*x,τ*) is the concentration of «free calcium hydroxide» in concrete at time *τ* [s] at an arbitrary point with the *x* coordinate [m], in terms of CaO, [kg CaO/kg concrete]; *k* is the mass conductivity coefficient, [m^2^/s]; *q_v_*(*x*) is the source of the mass of the substance as a result of a chemical reaction, [kg/(m^3^·s)]; *ρ_con_* is concrete density, [kg/m^3^]; *δ* is the wall thickness of the structure, [m]; *C*_0_(*x*) is the concentration of «free calcium hydroxide» in concrete at the initial moment of time at an arbitrary point with the *x* coordinate, [kg CaO/kg concrete]; *q_n_* is the density of the mass flow of a substance from concrete into a liquid medium, [kg/(m^2^·s)].

The initial condition (Equation (18)) shows that at the time taken as the starting point, the concentration of the transferred component («free calcium hydroxide») has a distribution over the thickness of the concrete structure.

The boundary condition (Equation (19)) is a condition of non-penetration at the outer boundary of the structure.

The boundary condition (Equation (20)) is a condition of the second kind and shows that there is a mass transfer between the solid and liquid phases at the boundary of the structure with a liquid medium.

In the final form, the boundary value problem of mass conductivity in dimensionless variables is written:(21)$$\begin{array}{c} \theta \left(\bar{x},Fo_{m}\right)=-\frac{Ki_{m}^{*}}{6}\left(6Fo_{m}+3{\bar{x}}^{2}-1\right)\\ +\frac{2Ki_{m}^{*}}{\pi^{2}}\overset{\infty }{\underset{n=1}{\sum }}\frac{{\left(-1\right)}^{n}}{n^{2}}\cos\left(\pi n\bar{x}\right)\cdot exp\left(-\pi^{2}n^{2}Fo_{m}\right)+\overset{1}{\underset{0}{\int }}\theta_{0}\left(\xi \right)d\xi \\ +2\overset{\infty }{\underset{n=1}{\sum }}\cos\left(\pi n\bar{x}\right)\overset{1}{\underset{0}{\int }}\theta_{0}\left(\xi \right)\cos\left(\pi n\xi \right)d\xi \cdot exp\left(-\pi^{2}n^{2}Fo_{m}\right)\\ \;\;\;\;\;\;+\left(Fo_{m}+\frac{3{\bar{x}}^{2}+2}{6}\right)\overset{1}{\underset{0}{\int }}Po_{m}^{*}\left(\xi \right)d\xi \\ \;\;\;\;\;\;-\overset{1}{\underset{0}{\int }}Po_{m}^{*}\left(\xi \right)\cdot \xi \cdot d\xi +\frac{1}{2}\overset{1}{\underset{0}{\int }}Po_{m}^{*}\left(\xi \right)\cdot \xi^{2}\cdot d\xi \\ \;\;\;\;\;\;-\frac{2}{\pi^{2}}\overset{\infty }{\underset{n=1}{\sum }}\frac{1}{n^{2}}\cos\left(\pi n\bar{x}\right)\overset{1}{\underset{0}{\int }}Po_{m}^{*}\left(\xi \right)\cos\left(\pi n\xi \right)d\xi \\ \;\;\;\;\;\;\cdot exp\left(-\pi^{2}n^{2}Fo_{m}\right), \end{array}$$
where $\theta \left(\bar{x},Fo_{m}\right)=\frac{C\left(x,\tau \right)-C_{0}}{C_{0}}$ is the dimensionless concentration; $\bar{x}=\frac{x}{\delta }$ is the dimensionless coordinate; $Fo_{m}=\frac{k\tau }{\delta^{2}}$ is the Fourier mass transfer criterion; $Ki_{m}^{*}=\frac{q_{n}\delta }{kC_{0}\rho_{con}}$ is the modified Kirpichev criterion; *n* is the number of row members; *ξ* is the coordinate of integration in the range $0\leq \xi \leq \bar{x}$; $Po^{*}\left(\bar{x}\right)=\frac{q_{v}\left(x\right)\delta^{2}}{kC_{0}\rho_{con}}$ is the modified Pomerantsev criterion.

## 3. Results and Discussion

Images of the cement stone surface (Figure 1) were obtained using a digital portable electron microscope USB Digital Microscope at a magnification of 50 times. The images show white crystals of calcium stearate in the pores and structure of cement stone. It can be noticed that with an increase in the concentration of the additive, the particle size becomes larger. Figure 1 shows that calcium stearate fills the pores, preventing the ingress of an aggressive environment deep into the cement stone. The study of the introduction of calcium stearate into the cement stone of concrete and aspects of pore colmatation can be carried out additionally and presented in subsequent publications.

During the setting period of the cement dough (from 0 to 2 h), moisture evaporation occurs faster in cement samples without additives than in hydrophobized ones (Figure 2). The beginning of the setting occurs after 1 h of holding the samples in the air and ends after 2 h from the moment of pouring into the mold. In samples containing calcium stearate additives, the onset of setting occurs after 2 h and ends 3 h after manufacture.

The results of the studies show that in samples with a concentration of calcium stearate of 1.1%, the mass change at the initial stage of hardening occurs uniformly. During the first 12 h, in samples with a hydrophobic additive content of 0.8% and 1.3%, the mass change occurs most intensively, while the concentration of the additive in the amount of 1.1% does not significantly affect the change in the mass of the samples compared to the sample without additives.

A greater change in the mass of samples with the addition of 0.8% and 1.3% calcium stearate may be due to the fact that the particles of the hydrophobizer slow down the interaction of the cement mixture with water, as a result of which the water remains on the surface and does not settle but evaporates. The difference in the intensity of the change in the mass of cement stone samples with hydrophobic additives during hardening may be due to the different distribution of calcium stearate particles in the cement volume during mixing.

However, after a day, the trend of changing the mass of samples does not persist, as can be seen in Figure 3. With further hardening, cement stone without the addition of calcium stearate shows a greater change in mass due to drying and evaporation of moisture not only from the surface but also from the structure in the subsurface layer. In hydrophobized samples, water remains trapped inside the pores due to the distribution of hydrophobizer particles in the cement stone structure [69,70,71]. Evaporation of moisture, in this case, occurs only on the surface of the samples, and the processes of hydration of cement continue inside [72]. This leads to the formation of more calcium-containing phases in the structure of cement stone, in particular calcium hydrosilicates, which are carriers of the mechanical strength of concrete [70,73,74,75]. As a result, hydrophobic concrete has improved strength characteristics [40,46,76].

Figure 3 shows that the change in the mass of cement stone samples with additives in the amount of 1.1% and 1.3% reaches constant values within 13–14 days; the samples showed the smallest change in mass during air hardening. In samples with a concentration of 0.8% calcium stearate additive, mass transfer processes slow down after 15 days of testing. Drying of the sample without additives takes place on the 18th day of exposure to air while achieving the maximum change in mass for the studied objects.

It is known that hydrophobic additives introduced into concrete at the manufacturing stage contribute to improving its operational characteristics [53,54,55,60]. The introduction of hydrophobic additives into the cement mixture reduces the amount of water evaporating during hardening, which leads to increased hydration processes and, as a consequence, changes in the characteristics of the cement stone of concrete (Table 3).

An increase in density due to a decrease in porosity should help slow down the penetration of an aggressive medium deep into the cement stone of concrete and inhibit the development of corrosion processes.

Determination of the kinetics of dissolution and leaching of calcium ions from cement stone by an aggressive chloride-containing medium was carried out when samples were immersed in a 2% MgCl_2_ solution. The content of calcium cations in the liquid medium was determined by the method described in Section 2.6. During the tests, it was found that the state close to the equilibrium concentration of calcium cations in the solution is reached after 14 days of stay of hydrophobized cement stone samples in a corrosive environment (curves 2, 3, and 4 in Figure 4) [58,77], whereas, for samples without a hydrophobizing additive, the equilibrium state occurs after 150 days of testing (curve 1 on Figure 4).

It is obvious that for cement concrete containing 1.1 and 1.3% by weight of the hydrophobizer, the kinetics of calcium leaching in an aggressive environment is almost the same (curves 3 and 4 in Figure 4); therefore, further studies were carried out on the samples of the W10 and W16 waterproof grades.

With the help of a derivatographic analysis, the content of calcium hydroxide in different parts of the cement stone was determined, which made it possible to calculate and construct profiles of the concentrations of «free calcium hydroxide» according to the thickness of hydrophobized cement stone samples exposed to a 2% MgCl_2_ solution at different stages of the experiment using the mathematical model (Equation (21)) (Figure 5 and Figure 6).

According to formulas (Equations (22) and (23)), the characteristics of the mass transfer process of “free calcium hydroxide” during the corrosion of hydrophobic concrete in 2% MgCl_2_ solution were calculated (Figure 7 and Figure 8).
(22)$$k=\frac{q}{\rho_{0}\frac{dC}{dx}},\;$$
where *k* is the mass conductivity coefficient, [m^2^/s]; *q* is the mass flow density due to chemical reactions, [kg/(m^2^·s)]; *ρ*_0_ is the density of the solid phase, [kg/m^3^].
(23)$$q=\frac{\Delta C_{liq}}{S\cdot \tau },\;$$
where Δ*C_liq_* is the mass of the substance transferred from the cement stone to the liquid medium, [kg]; *S* is the area of the corroded concrete surface, [m^2^]; *τ* is the process time, [s].

To calculate the processes of substance transfer, it is convenient to enter the mass transfer coefficient *β*. It is defined as the ratio of the diffusion flux *q* to the concentration difference ∆*C*, [kg/m^3^]:(24)$$\beta =\frac{q}{\Delta C}.\;$$

Figure 8 shows that the mass conductivity coefficient of hydrophobized cement stone samples under corrosion in a 2% MgCl_2_ solution varies exponentially. The curves have a flatter appearance compared to the curve of a non-hydrophobized sample, which indicates a significantly lower rate of corrosion processes in this case. The mass transfer coefficients behave in a similar way (Figure 8).

The shape of the curves in Figure 7 and Figure 8 (letter A) corresponds to the slowing down of mass transfer processes occurring in hydrophobized cement stone during liquid corrosion and is consistent with the kinetic curves shown in Figure 4. This behavior is caused by the inhibition of the dissolution of calcium-containing phases in cement stone under the influence of a liquid-aggressive medium [78] due to the complication of the penetration of chloride ions deep into the pore structure of cement stone due to the hydrophobicity of the pore surface and their colmatation with calcium stearate [58]. The values of the mass transfer coefficients corresponding to the equilibrium state are used to determine the terms of maintenance-free service life of the concrete and reinforced concrete products under the influence of liquid-aggressive media [79].

## 4. Conclusions

The processes of mass transfer and mass conductivity significantly slow down with an increase in the density of concrete. Consequently, due to the formation of a denser structure of cement stone, including through volumetric hydrophobization, the corrosion resistance of concrete in environments of varying degrees of aggressiveness increases.

In comparison with some other hydrophobic additives, calcium stearate does not hamper hydration processes during concrete hardening. The addition of stearate into the cement mixture in the manufacture of concrete contributes to an increase in strength due to the formation of a large number of calcium-containing phases in the structure due to a more complete interaction of cement components with non-evaporating water. Calcium stearate belongs to hazard class 3; that is, it is moderately dangerous. Since the concentration of additives in concrete does not exceed 1.5% by weight of cement, they have no impact on human health and the environment.

To ensure uniform volumetric hydrophobization of cement stone, it is necessary to introduce calcium stearate in powdered form followed by thorough mixing. The introduction of calcium stearate increases the density of concrete cement stone, as can be seen in Table 3, which makes it difficult to use it for the hydrophobization of light and air-entrained concretes.

The introduction of hydrophobic additives increases the service life of a concrete product in case of corrosion in a highly aggressive liquid chloride-containing medium by four times. Volumetric hydrophobization allows for reducing water absorption and increasing the density of cement stone, which means reducing the amount of aggressive medium entering the concrete and reducing the degree of corrosion destruction of cement stone. The conducted studies of the corrosion resistance of cement stone with hydrophobizing additives in a liquid chloride-containing medium using mathematical modeling allowed us to determine the parameters of mass transfer (coefficients of mass conductivity and mass transfer), which can be used to establish the terms of safe operation of concrete structures.

The paper defines the main parameters of corrosion mass transfer (coefficients of mass conductivity and mass transfer) for Portland cement of the CEM I 42.5 N brand with the addition of a calcium stearate hydrophobizer in a highly aggressive medium, which can be used to calculate the service life of concrete and reinforced concrete products using mathematical models of corrosion.

The obtained ideas about the corrosion destruction of hydrophobized concrete, taking into account the regularities of mass transfer processes, allow us to predict the effects of liquid media of varying degrees of aggressiveness on concrete and develop recommendations for improving the corrosion resistance of concrete and reinforced concrete products.

## Figures and Tables

**Figure 1 materials-16-03827-f001:**
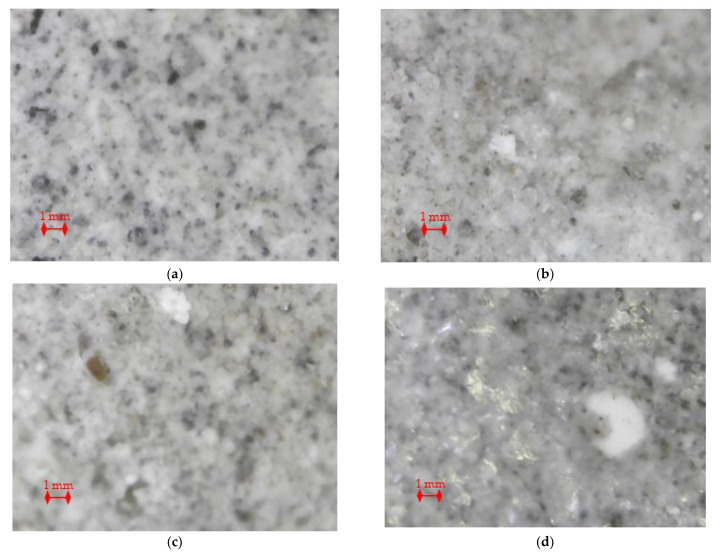
Images of the surface of concrete samples: (**a**) without calcium stearate; (**b**) the concentration of the hydrophobic additive is 0.8%; (**c**) the concentration of the hydrophobic additive is 1.1%; (**d**) the concentration of the hydrophobic additive is 1.3%.

**Figure 2 materials-16-03827-f002:**
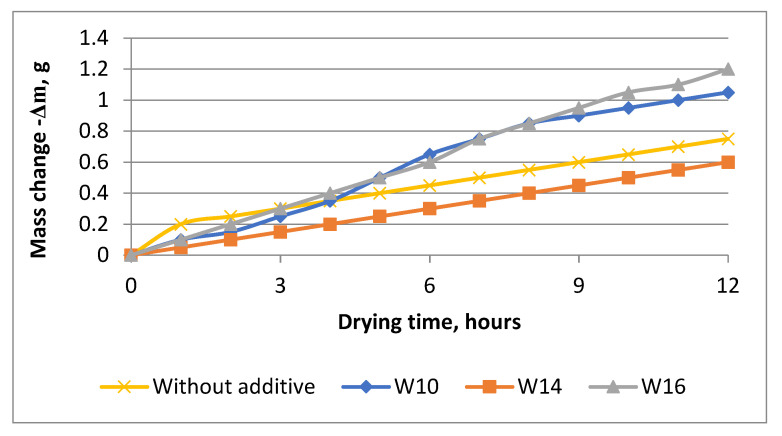
Change in the mass of cement stone samples with calcium stearate additives during the first 12 h of air hardening.

**Figure 3 materials-16-03827-f003:**
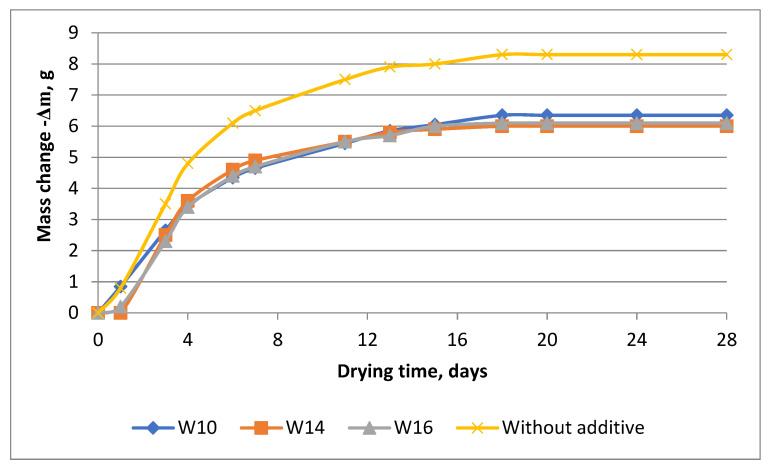
Change in the mass of cement stone samples with calcium stearate additives during air hardening at 20 °C.

**Figure 4 materials-16-03827-f004:**
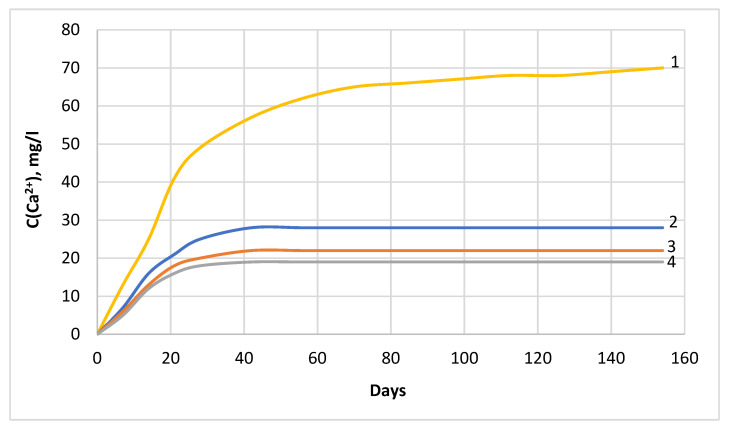
Kinetic curves of the concentration of calcium cations in a 2% MgCl_2_ solution of cement stone samples: (1) non-hydrophobic; (2) waterproof grade W10; (3) waterproof grade W14; (4) waterproof grade W16.

**Figure 5 materials-16-03827-f005:**
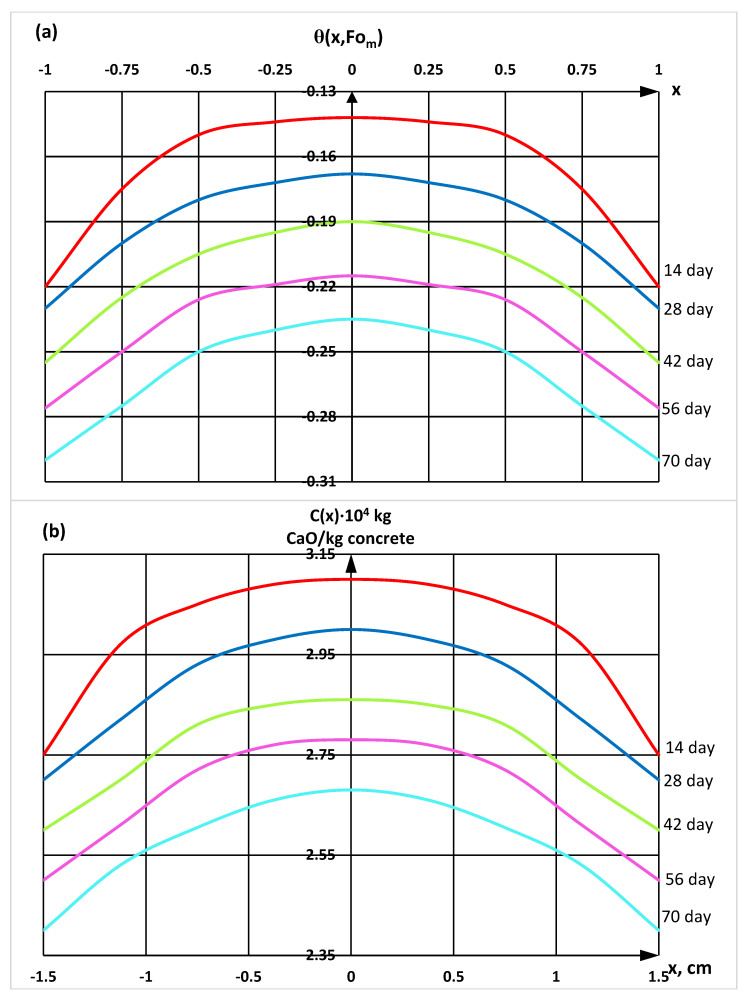
Profiles of concentrations of «free calcium hydroxide» by the thickness of the W10 waterproof grade sample in dimensionless coordinates (**a**) and in coordinates with real physical dimensions (**b**).

**Figure 6 materials-16-03827-f006:**
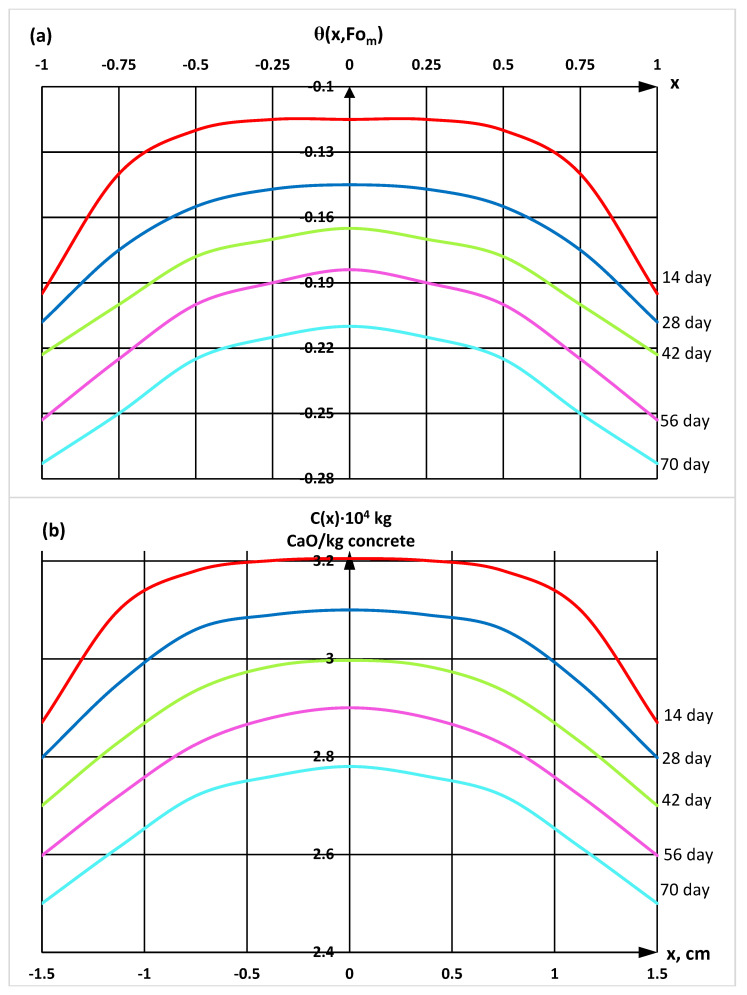
Profiles of concentrations of «free calcium hydroxide» by the thickness of the W16 waterproof grade sample in dimensionless coordinates (**a**) and in coordinates with real physical dimensions (**b**).

**Figure 7 materials-16-03827-f007:**
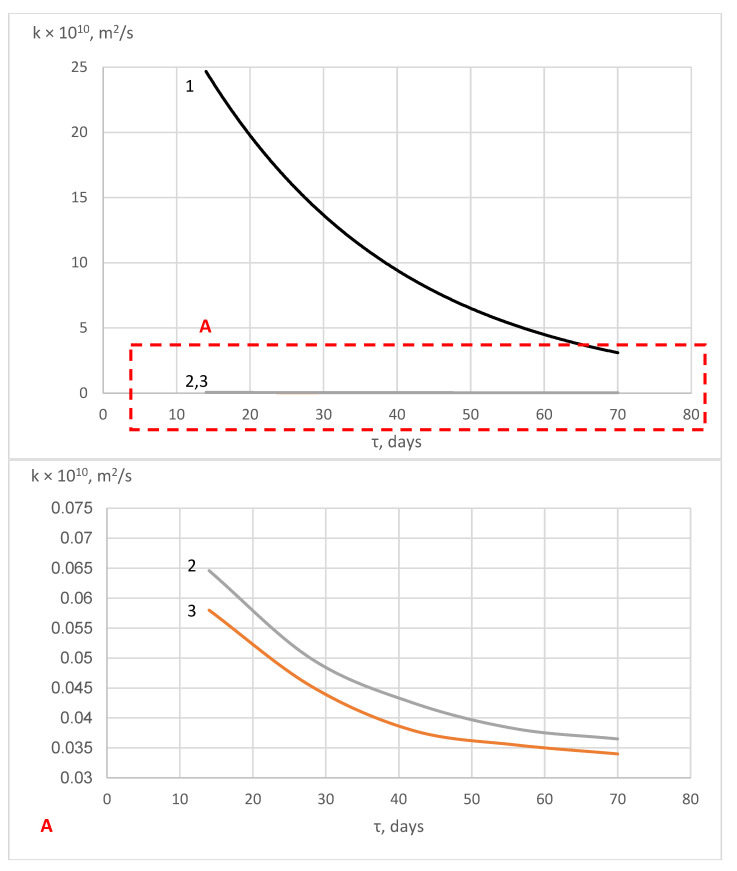
Change in the values of the mass conductivity coefficient during corrosion of cement stone samples in a 2% MgCl_2_ solution: (1) sample without additive; (2) sample of the W10 waterproof grade; (3) sample of the W16 waterproof grade.

**Figure 8 materials-16-03827-f008:**
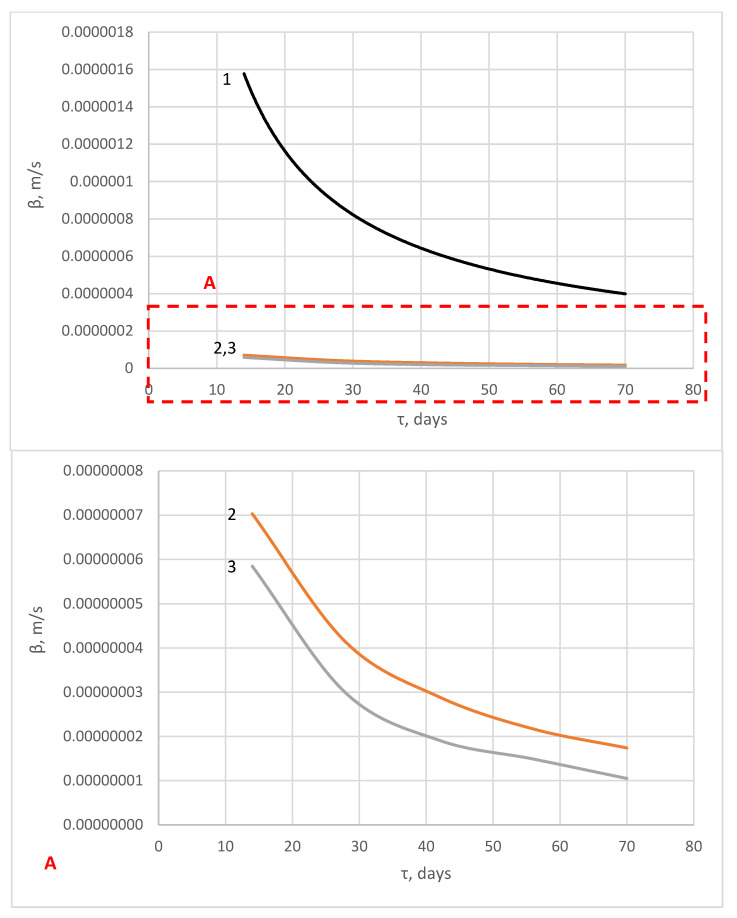
Change in the values of the mass transfer coefficient during corrosion of cement stone samples in a 2% MgCl_2_ solution: (1) sample without additive; (2) sample of the W10 waterproof grade; (3) sample of the W16 waterproof grade.

**Table 1 materials-16-03827-t001:** Chemical composition of Portland cement CEM I 42.5 N.

Components	SiO_2_	Al_2_O_3_	Fe_2_O_3_	CaO	MgO	SO_3_	R_2_O
Amount [%]	21.24	5.65	4.30	65.87	0.86	0.44	0.71

**Table 2 materials-16-03827-t002:** Mineralogical composition of Portland cement CEM I 42.5 N.

Components	C_3_S	C_2_S	C_3_A	C_3_AF
Amount [%]	61.40	14.60	7.69	13.01

**Table 3 materials-16-03827-t003:** Characteristics of hydrophobized cement stone.

Characteristic	Concrete Grade for Water Resistance		
	W10	W14	W16
Density [kg/m^3^]	2432.1	2568.2	2604.4
Water absorption [%]	4.0	3.7	3.5
Porosity [%]	6.8	6.0	5.7
Strength [MPa]	54.58	58.18	63.39

## Data Availability

Data sharing is not applicable to this article.

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
