# Peer review of "Investigation of the Effect of Volumetric Hydrophobization on the Kinetics of Mass Transfer Processes Occurring in Cement Concretes during Corrosion"

_materials, 2023, doi:10.3390/ma16103827_

Round 1
Reviewer 1 Report
This manuscript entitled “Investigation of the Effect of Volumetric Hydrophobization on the Kinetics of Mass Transfer Processes Occurring in Cement Concretes during Corrosion” focuses on the effect of hydrophobic additive of calcium stearate into the cement mixture at the stage of concrete production. It is potentially of interest to the readership, however, the research results reported are too premature for publication.
- Figure 2 and its caption are too premature for publication.
- Figure 9 should use the same numerical values as the y axis.
Author Response
Dear reviewer, could you provide a detailed explanation of the remark "too premature for publication" in your comments?
The values on the y axis in Figure 9 are corrected.
Reviewer 2 Report
In my view, the authors have conducted a very interesting work which has good impact to the field. Regardless, the authors should carefully address the comments shown below:
(1) With regards to Figure 1: if the authors drew the schematic themselves, it is fine, however, if they took it from somewhere this needs to be cited properly. Please also improve the quality of the figure.
(2) (i) More detailed comparison/analysis should be conducted for the images of the surface of concrete samples shown in Figure 2. (ii) How did the authors capture the images? Please provide more info. (iii) In the figure caption, (a) please fix the language, it looks like it is written in Russian.
(3) What dictates the concentration of additives used in this study? Why this range was selected?
(4) Figure 4: Please use symbols for the data corresponding to "no additives".
(5) (i) Can the authors conduct sensitivity analysis to see how the concentrations of «free calcium hydroxide» shown in Figure 5 changes? (ii) What are the parameters used to obtain the data? It would be good to have them summarized in a Table.
(6) State the novelty of the work clearly and provide recommendations for future research directions.
(7) Improve the figure quality and proofread the manuscript to ensure there are no grammatical as well as typographical errors.
(8) The authors need to improve the literature review. There are studies that highlight the importance of :(i) rheology on microstructure of material and (ii) sensitivity analysis to extract important kinetic parameters:
1. https://doi.org/10.1139/cjc-2021-0248
2. https://doi.org/10.1016/j.porgcoat.2022.107172
3. https://doi.org/10.1016/j.rinma.2023.100373
Author Response
Dear reviewer, thank you for your comments. Changes have been made to the text of the article on most points, they are highlighted in yellow. Below are the answers to some of the comments:
(1) Figure 1 is a stock image from the Internet, such pictures have no authorship.
(5) An explanation is given to the test procedure in the text of the article. The content of calcium cations in the solution was determined by titration, it is impractical to present these data in the form of a table, since this will not give a visual changes, as shown in Figure 5.
(8) These articles are not related to the subject and objects of research of this work.
Reviewer 3 Report
The article is well established and includes good contributions
Author Response
Dear reviewer, thank you for your positive comment.
Reviewer 4 Report
The manuscript describes studies on the effect of protective calcium stearate layer on the concrete and the corrosion properties of the materials. The manuscript is well written, and the obtained results interesting, and show convincingly that calcium stearate can prevent corrosion in concrete material even in harsh MgCl2 environment. Furthermore, the author presents rather detailed mathematical analysis to support the conclusions. I have only a couple of very minor comments, which can be easily corrected before publication in Materials.
Table 1 and 2: Amount, not ammount.
Figure 1 is quite unnecessary since it presents very standard titration equipment.
Figure 2. Please translate the russian language into english in (a).
Author Response
Dear reviewer, thank you for your positive comment. Corrections according to your comments in the text are highlighted in green.
Reviewer 5 Report
In this manuscript, the effects of calcium stearate additive as a hydrophobic additive to prevent corrosion and destruction of cement stone structure by foreign matter entering the pores of concrete were investigated.
This manuscript may be accepted after major revision.
1. Table 3 should be given after Table 1 and Table 2 in Chapter 2 or it can be made into a single table.
2. Table 2.1 and Table 2.2 are missing in the text (line 98-99). Are the Table 2.1 and Table 2.2 Table 1 and Table 2?
3. According to the description of Table 6 in MI 2625-2000 on line 136, Table 6 is absent.
4. Information in line 141-145 is not included in Table 2. Table 6 is mentioned in Line 147, but Table 6 is not in the manuscript.
5. On Line 158-159, the sentence “where: γL is true density of coarse aggregate, [g/cm3 ]; К5 is stoichiometric constant from 158 Table 2 in MI 2625-2000” is written. However, this information is not available in Table 2.
6. In the studies carried out to determine the compressive strength of the cement stone, which model tool was used so some visuals should be given about the method.
7. What does the Q-1500D derivatograph on Line 232 mean?
8. Line 36 contains a Russian sentence.
9. Figure 1 is the shape of a very well known method. It does not need to be included in this study. Only literature can be given to the method.
10. The expressions (18), (19) and (20) on lines 274, 277 and 279 should be written as (equation 18), (equation 19) and (equation 20)
11. The method in Figure 2 should be defined (Line 289). Are the Figure 2 SEM photos? SEM photographs should be interpreted in more detail. The pores, voids and particle sizes on the surface should be given in detail and the effect of the calcium stearate additive used on the morphology should be discussed.
12. Line 291 has a Russian sentence.
13. The differences in sample mass change at 12 hours and 7 days between 1.10% and 1.30% in Figure 3 and Figure 4 should be explained based on examples in the more detailed literature. Figure 3 and Figure 4 interpretations should be developed.
14. Curves 1 and 2 in Figure 5 should be mentioned.
15. The mathematical model (21) on Line 343 is not found in 21 text.
16. Figure 7’ should be better drawn and clear.
17. Equality expression should be used instead of the expressions in 353 and it would be more appropriate to write it as (equation 22) and (equation 23).
18. Figures 8 and 9 should be explained in more detail and compared with studies in the literature.
As a result, in the results of this study, information should be given about how superior the hygroscopic material calcium stearate, which is used to reduce cement corrosion, is compared to the materials in the literature. It should be compared in detail with the results in the literature. However, I believe that it will be more beneficial to study by taking these evaluations into consideration.
Author Response
Dear reviewer, thank you for a thorough and useful review. The corrections made according to your comments in the text are highlighted in blue. The following answers are available to your comments:
- Amended.
- The numbering of the tables has been corrected.
- Table 6 referred to the document MI 2625-2000, and not to the text of the article. The mention of this table has been removed.
- Table 2 here referred to the MI 2625-2000 document, and not to the text of the article. The mention of this table has been removed.
- Table 2 here referred to the MI 2625-2000 document itself, and not to the text of the article. The mention of this table has been removed.
- In paragraph 2.4, an image and a diagram of the press on which the samples were tested for strength were added.
- Q-1500D is a brand of derivatograph.
- No Russian sentence was found in line 36.
- The scheme for titration has been removed.
- The corresponding changes have been made.
- Images of the cement stone surface were obtained using a digital portable electron microscope USB Digital Microscope. The information is added before Figure 2.
- The phrase has been translated.
- An addition has been made to the text of the article.
- The mention of curves 1 and 2 of Figure 5 is included in the text of the article.
- Equation 21 was meant here, not the source [21]. Correction has been made to the text of the article.
- Figures 6 and 7 have been changed.
- Corrections have been made.
- An addition has been made to the text of the article.
Round 2
Reviewer 1 Report
- The author should make sure the new figure 1 is original, rather than picking up online images.
- Figrue 2 should have any scale bar in the image to describe the size and scale of the morphology at all.
Author Response
Dear reviewer, according to your comments, changes have been made:
- Figure 1 has been removed because it is a standard test equipment, the principle of its operation is described in the text and is intuitive, and the appearance of the device does not affect the results.
- In the images of Figure 1 (previously Figure 2) the scale is applied.
Reviewer 2 Report
The authors failed to incorporate the changes I requested which were required to make this work in a publishable format. Unfortunately, this work does not meet the high requirements of the journal at this current stage.
Author Response
Dear reviewer, the attached file lists the changes made to the text of the article according to your comments. They are highlighted in yellow in the manuscript. Please specify which changes you consider insufficient?
